# Pertussis—A Re-Emerging Threat Despite Immunization: An Analysis of Vaccine Effectiveness and Antibiotic Resistance

**DOI:** 10.3390/ijms26199607

**Published:** 2025-10-01

**Authors:** Anna Duda-Madej, Jakub Łabaz, Ewa Topola, Hanna Bazan, Szymon Viscardi

**Affiliations:** 1Department of Microbiology, Faculty of Medicine, Wroclaw Medical University, Chałubińskiego 4, 50-368 Wrocław, Poland; 2Faculty of Medicine, Wroclaw Medical University, Ludwika Pasteura 1, 50-367 Wrocław, Poland; jakub.labaz@student.umw.edu.pl (J.Ł.); ewa.topola@student.umw.edu.pl (E.T.); hanna.bazan@student.umw.edu.pl (H.B.); szymon.viscardi@student.umw.edu.pl (S.V.)

**Keywords:** acellular vaccines, *Bordetella pertussis*, hemagglutinin, pertactin, pertussis, pertussis toxin, whole-cell vaccines, whooping cough

## Abstract

Pertussis is an infectious disease that contributes to hundreds of thousands of deaths worldwide each year. Despite the prevalence of preventive vaccination programs, there has been an increasing number of new cases of the disease over the past few decades. This poses a particular problem for the pediatric population among whom the highest mortality from the disease is recorded. Several reasons for this phenomenon can be mentioned, but what is particularly important from the microbiological point of view is the correlation of the increased number of pertussis cases with the introduction of a new form of vaccine—the acellular vaccine in place of the whole-cell vaccine. In this review, we summarized the current state of knowledge on potential factors that may contribute to the decline in immunization efficacy against the pathogen. The post-vaccination response profile, symptomatic of vaccination with vaccination-acellular, is characterized by recruitment of Th_2_ and Th_17_ lymphocytes; it has been reported that in the long term, this results in insufficient activation of B cells and low titers of antibodies to key bacterial antigens (hemagglutinin, pertactin). Moreover, the immune response proceeds by bypassing the recruitment of tissue-resident memory T cells, resulting in a lack of protection against colonization of the nasal cavity by the bacterium despite vaccination. The decline in vaccination efficacy should also be attributed to the phenotypic variability of *Bordetella*. The popularization of the PtxP3 strain, characterized by its ability to incompletely activate immune mechanisms, poses a real threat to public health. The growing resistance of *B. pertussis* to standardly used antibiotics including macrolides also remains a problem. This makes it difficult to eradicate pathogens from the nasal cavity area and increases the pool of bacterial carriers in the population area. The increasing prevalence of the disease prompts reflection on more effective methods of prevention. Particularly promising in this field seem to be new vaccines, especially mucosally implemented, e.g., intranasal, or developed on the basis of *B. pertussis* antigens other than those used so far.

## 1. Introduction

Pertussis, also known as whooping cough, was first described during the Paris epidemic in 1578. This disease is caused by *Bordetella pertussis*—a small, aerobic Gram-negative bacterium belonging to the *Alcaligenceae* family, with a shape intermediate between cocci and bacilli. It was first isolated in pure culture in 1906 by Bordet and Gengou [1,2,3].

Humans are the only reservoir of the *B. pertussis*, which rapidly colonizes the ciliated epithelium of the respiratory tract. The transmission of the bacteria occurs via airborne droplets produced during coughing or through direct contact with a sick person [1,4,5]. The infectious dose of *B. pertussis* is small—the disease develops in about 80% of people who have had contact with the pathogen, which makes whooping cough one of the most contagious diseases [6,7]. Whooping cough is also caused by other species of the Bordetella genus, mainly by *Bordetella parapertussis*, which also colonizes the human respiratory tract. However, it causes a milder course of the disease, which is why its infection is called parapertussis [2,3]. *B. pertussis* infection presents as an acute respiratory infection, affecting mainly children under 5 years of age, in whom it manifests itself with a long-lasting cough of a paroxysmal and spastic nature with the secretion of viscous sputum [5,8]. In adults, the course of the disease is often asymptomatic. Nevertheless, if symptoms do appear, they are not very specific and manifest mainly in the form of a chronic, non-characteristic cough [9]. It can persist for up to 3 months; hence, another common name for pertussis is “100-day cough” [10].

Pertussis is one of the most common infectious diseases causing death in children [11]. The mortality rate in this age group is about 2%, which accounts for 96% of deaths related to pertussis in all age groups [5]. According to WHO data from 2014, 50 million cases of whooping cough are recorded each year, 300 thousand of which end in death [12]. These incidences are on an upward trend, as indicated by data from most countries [13].

For example, in Germany, according to data from the Robert Koch-Institut (RKI), 25,271 pertussis cases were officially recorded in 2024 compared to 3429 cases in the preceding year—indicating a more than sevenfold increase and the highest incidence since the introduction of national mandatory reporting in 2013 [14].

The high infectivity of *B. pertussis* is closely linked to its virulence, which is regulated by the BvgAS two-component system, allowing adaptation to environmental cues [1,2]. It relies on a coordinated expression of virulence factors, primarily adhesins and toxins. Adhesins (e.g., fimbriae FIM2 and FIM3, filamentous, hemagglutinin (FHA) and pertactin (PRN)) enable colonization of the respiratory epithelium. In contrast, toxins (e.g., pertussis toxin (Ptx); adenylate cyclase toxin (ACT), lipooligosaccharide (LOS), tracheal cytotoxin (TCT), and dermonecrotic toxin (DNT)) play roles in immune evasion, inflammation, and epithelial damage [1,15].

Figure 1 presents a simplified diagram of the pathogenesis of *B. pertussis* infection.

The infection occurs through droplets, *B. pertussis* colonizes the tracheal epithelium (cilliated cells (1), goblet cells (2), basal cells (3)) using specific adhesins: FHA, PRN, and fimbriae (I). Bacteria induce a response from TLR (toll-like receptors), developing an initial immune response. During this time, they are also absorbed by dendritic cells (II), which activate T lymphocytes in the lymph nodes, which differentiate towards Th_17_ (III). These lymphocytes, via secreted cytokines, activate and cause the migration of neutrophils to the site of infection (IV). During inflammation, numerous antipathogenic factors and reactive oxygen species (ROS) are secreted (V). Bacteria produce numerous exotoxins (e.g., pertussis toxin, cytotoxin), which, together with inflammatory factors, cause massive destruction of the tracheal epithelium (VI).

In order to protect against pertussis, common and, in many countries also, mandatory vaccination of children was introduced (Poland—1960; German Democratic Republic—1964, Federal Republic of Germany—1969; Great Britain—1950s, USA—registration of the first whole-cell pertussis vaccine in 1914; DTP commonly used in most states since the 1980s). Currently, the vaccination schedules of most of the above-mentioned countries use the acellular form of the pertussis vaccine, primarily as a combination vaccine with *C. diphtheriae* and *C. tetani* antigens (DTaP) [16,17,18]. The only exception is Poland, where up to 18 months of age, four doses of the whole-cell vaccine (DTP) are administered, unless they are contraindicated (if they occur, then the acellular form is administered). Subsequent vaccinations for Polish children are based on the acellular vaccine (DTaP, Tdap) [19].

The aim of the pertussis vaccination is to resolve long-term T cell responses. Immunization was designed to activate, primarily, Th_1_ and Th_17_ cells, which play a critical role in clearing pathogens from the respiratory mucosa and limiting *B. pertussis* transmission. Whole-cell vaccines (wP) promote immune polarization toward Th_1_ and Th_17_ subsets, while acellular vaccines (aP) favor a Th2 response, which is more effective at reducing carriage rather than preventing infection [20]. Moreover, aP vaccines elicit limited cellular immunity (mainly enhance the synthesis of antibodies) and only modest Th_1_ activation, thereby offering suboptimal long-term protection against *B. pertussis* [20,21]. Consequently, an effective vaccination strategy that ensures both infection control and environmental bacterial reduction should ideally include both wP and aP components.

Despite the calm for many years, recently there has been an upward trend in cases with cyclical waves every 3–5 years, with a peak in the summer months [22,23]. The immunization strategy is becoming less effective due to (i) waning immunity despite completion of the full vaccination schedule; (ii) the reduced long-term efficacy of aP; (iii) the circulation of mutated *B. pertussis* strains; and (iv) the emergence of strains lacking expression of vaccine antigens [24]. It is also worth noting that a clear correlation with protection against *B. pertussis* has not yet been established, although elevated levels of some pertussis-specific antibodies (e.g., anti-PT-IgG, anti-PRN) are associated with a reduced risk of the disease. However, studies show that antibody titers decline over time, and IgG-PT alone is not sufficient to predict protection. This uncertainty likely contributes to occasional vaccination failures and the need for booster doses [25,26]. All these factors have contributed to declining post-immunization protection and loss of defense against infection, resulting in continued bacterial circulation in the population and the emergence of oligosymptomatic and asymptomatic infections [4,27].

For the reasons mentioned above, it follows that neither vaccination in childhood nor recovering from the disease guarantees lifelong protection. Therefore, it is possible and quite common to be infected with pertussis several times in one’s life; unless a booster dose of the vaccine is administered every 5 years, which allows pertussis-specific antibodies and cellular immunity to remain above pre-booster levels [28,29]. The study by Gao et al. [30] provided us with compelling evidence that the humoral response produced after a received vaccination (aP) declines rapidly. They proved that it had antibodies against FHA and PRN decreased gradually with time after vaccination, while anti-Ptx antibodies remained constant. Therefore, meta-analyses conducted showed that the probability of pertussis incidence increased up to 8-fold at 8–9 years after vaccine injection [30]. In addition, the non-characteristic course of the disease in adults makes diagnosis difficult and delays the implementation of treatment, which may result in the spread of *B. pertussis* among society [5,9].

Although the introduction of acellular pertussis (aP) vaccines is often cited as a contributing factor to the resurgence of pertussis, other factors should also be noted. The aforementioned asymptomatic course of disease, often occurring in adults, plays an important role, making them a difficult-to-detect source of pertussis spread [9]. Furthermore, improvements in diagnostic methods (such as the increased use of PCR), greater clinician awareness, and lower testing thresholds can also been identified as contributing factors to the increase in reported cases [31]. Continuous changes in vaccination schedules, including the timing and number of booster doses administered, are also important, as demonstrated by modeling and a serological study by Paireau et al. in France in 2013, following a change in the childhood vaccination schedule in that country [32].

In the presented article, we will discuss the response from the immune system to the vaccines used and the antigens contained in them that provide “protection” against pertussis. We investigated more closely the problem resulting from adaptive changes in *B. pertussis*, affecting the decline in vaccine efficacy. To the best of our knowledge, this is the first review article approaching the existing problem from the side of bacterial variability. Through our article, we wanted to remind us of a bacterium that has been “forgotten” for many years and to emphasize the need for continued research into the genome of this pathogen susceptible to constant change.

## 2. Methods

In this review, we searched for articles by using the databases Scopus, PubMed, Web of Science, and Google Scholar. In total, 202 articles were cited. The articles were filtered by searching for the following keywords in the title and abstract of the articles: ”pertussis vaccine”, “acellular vaccine”, “whole-cell vaccine”. Only articles written in English were qualified for this review. 

## 3. Virulence Factors of *Bordetella pertussis*

***Initiating stage:*** *B. pertussis* initiates respiratory infection through a suite of adhesins that mediate attachment to ciliated epithelial cells [4]. The key ones among these are FHA—a multifunctional adhesin that both enables strong binding to airway mucosa and modulates local immune responses—and fimbriae (FIM2/FIM3), which facilitate colonization of the nasopharynx [1]. On the other hand, PRN, another surface adhesin, contains Arg-Gly-Asp motifs that bind *B. pertussis* to receptors located on the surface of mammalian cells [33]. Loss-of-function mutations in these adhesins markedly impair bacterial colonization, highlighting their critical roles during the early stages of pertussis pathogenesis [34].

***Toxins modulating host signaling and immunity:*** The primary and the best characterized virulence factor of *B. pertussis* is Ptx, produced exclusively during bacterial growth. The enzymatically active S1 subunit functions as an ADP-ribosyltransferase, inactivating the Gi protein and leading to elevated intracellular cAMP levels [35]. This disrupts numerous host signaling pathways, resulting in (i) enhanced susceptibility to anaphylaxis (via sensitization to histamine and serotonin) [1,36]; (ii) inhibition of epinephrine-induced hyperglycemia (through increased insulin secretion) [37]; (iii) immune modulation (increased leukocyte migration and lymphocytosis) [38,39], and (iv) excessive mucus production in the bronchial epithelium [40,41]. Figure 2 shows the pattern of action of Ptx and the resulting immunological consequences.

Adenylate cyclase toxin (ACT) also increases intracellular cAMP, impairing the microbicidal and cytotoxic functions of leukocytes, including monocytes, neutrophils, and natural killer (NK) [1,39].

On the other hand, BrkA (*Bordetella* resistance to killing A) is a surface-associated protein that protects *B. pertussis* from complement-mediated lysis, enhancing bacterial survival within the host [2].

***Endotoxins and inflammatory mediators:*** *B. pertussis* produces a heat-stable endotoxin in the form of lipooligosaccharide (LOS), functionally similar to lipopolysaccharides (LPS) of other Gram-negative bacteria. Classified as agglutinogen 1 (AGG1), LOS lacks the long O-antigen side chain typical of LPS molecules from related species. LOS by triggering toll-like receptor 4 (TLR4) contributes to the inflammatory response during early infection by stimulating the production of interleukin-1 (IL-1), interleukin-8 (IL-8), and tumor necrosis factor-alpha (TNF-α), leading to fever, hypotension, and systemic inflammation [1,42].

***Cytotoxins damage respiratory epithelium:*** Tracheal cytotoxin (TCT) is unique among *B. pertussis* virulence factors in that its expression is not environmentally regulated. TCT directly damages ciliated epithelial cells, likely through the induction of nitric oxide and/or IL-1α, which inhibit DNA synthesis and repair cell regeneration. The loss of ciliary function leads to defective mucociliary clearance and is a principal cause of the paroxysmal cough characteristic of pertussis [1,2].

*B. pertussis* also produces a heat-labile dermonecrotic toxin (DNT), although its role in human infection remains unclear. In murine models, subcutaneous administration of DNT induces localized skin necrosis, suggesting cytotoxic and vasoconstrictive properties. It is hypothesized that DNT, in combination with TCT, contributes to respiratory epithelial damage through localized ischemia [1,2,39].

## 4. Clinical Manifestation of Pertussis

The clinical presentation of pertussis varies with patient’s age, sex, as well as immunization and vaccination status. The disease course is classically divided into four phases: incubation, catarrhal, paroxysmal, and convalescent [3,5,43].

***Incubation phase:*** The incubation period lasts between 6 and 20 days (mean: 7 days) and represents the time from exposure to *B. pertussis* to the onset of initial symptoms. This phase is asymptomatic [44].

***Catarrhal phase:*** Lasting approximately 1–2 weeks, this phase begins with nonspecific upper respiratory symptoms resembling a mild viral infection: rhinorrhea, pharyngitis, fatigue, and conjunctival infection. Fever is typically absent or low-grade. A dry, mainly nocturnal cough may be paroxysmal and occur first at night and then also during the day [5,45]. This phase is characterized by high infectivity, as the patient releases large quantities of *B. pertussis* through respiratory secretions [39].

***Paroxysmal phase:*** This stage typically lasts 2–6 weeks and is characterized by intense paroxysms of spasmodic coughing, often ending in a high-pitched inspiratory “whoop”, particularly in children—a hallmark of pertussis. In adults, this feature may be absent. Coughing episodes may be accompanied by dyspnoea, cyanosis, facial enema, post-tussive emesis, fainting, diaphoresis, and fatigue. If fever is present, it usually resolves by this phase [5,8]. Hematological findings include marked leucocytosis (≥100.000/mm^3^), with 60–80% lymphocytes, sometimes mimicking leukemia [39]. The patient may remain infectious for up to 3 weeks after cough onset [46].

***Convalescent phase:*** The recovery phase may last for weeks to months, with a gradually diminishing cough [44,47]. Residual cough is attributed to persistent epithelial damage and delayed tissue regeneration due to cytotoxin activity. Impaired ciliary function predisposes the respiratory tract to secondary infections and irritant-induced coughing [5,39]. In children, auscultation may reveal inspiratory wheezes or rales, potentially indicating secondary pneumonia. In contrast, infants are at risk of tachypnoea, apnoea, and episodic bradycardia [5].

Common complications of whooping cough in children include weight loss, dehydration, fainting, insomnia, otitis media. However, the most dangerous consequence of pertussis is damage to the respiratory system and central nervous system [43]. Respiratory complications during the course of pertussis are the result of an excessive immune response to the presence of bacteria in the ciliated cells lining respiratory tract (trachea, bronchi and bronchioles) as well as in the pulmonary macrophages [2]. Secondary bronchopneumonia with partial atelectasis often occurs and is caused by aspiration of gastric contents during coughing paroxysms as well as reduced clearance of pathogens from the respiratory tract due to damaged epithelium [5]. In extreme cases, respiratory complications can lead to apnoea and, consequently, to death [43]. On the other hand, the main complication affecting the nervous system is acute encephalopathy. It can result from cerebral hypoxia, toxic damage to this organ, and intracranial bleeding caused by severe paroxysmal coughing. Symptoms of damage to the central nervous system include impaired consciousness, epileptic seizures, and convulsions, even leading to death [2,43]. According to the Centers for Disease Control and Prevention (CDC), encephalopathy results in in 0.4% of children under 12 months of age being hospitalized due to pertussis, of which about 1/3 develop irreversible brain damage [2].

In older children and adults, complications are primarily a consequence of intense coughing rather than the action of bacterial toxins, and may manifest themselves as rib pain, hernias (mainly inguinal), conjunctival hemorrhages, and facial ecchymosis [2,43]. The risk of severe course of disease and the occurrence of complications can be minimized by the development of a full immune response to *B. pertussis* as a result of complete vaccination as well as by early diagnosis of infection and appropriate antibiotic therapy implemented as soon as the first symptoms appear] [48,49]. Antibiotics used in the treatment of *B. pertussis* infection are primarily macrolides (azithromycin, clarithromycin, and erythromycin), and also trimethoprim–sulfamethoxazole [48].

## 5. Types of Vaccines

The most efficient way of protection against pertussis is vaccination. Two types of vaccines can be distinguished—wP and aP [50]. The first one comprises inactivated whole bacteria, whereas the second one comprises purified components of it. All of these are administered intramuscularly, although research is currently underway on a new method of administration: intranasal delivery [51]. The described distinction is crucial as it influences many things like f.e. side effects correlated with an administered vaccine. wP and aP vaccines can be compared in terms of efficacy and effectiveness, duration of protection, reactogenicity, and cost [52].

Due to their heterogenous nature, wP vaccines provide more effective protection against pertussis compared to aP vaccines, which contain only few *B. pertussis*’s proteins. It is connected with the ability of wP vaccine to prevent not only the emergence of the symptoms but also the colonization of the respiratory tract [53]. This can improve herd immunity, which is a definite advantage of wP vaccines over aP ones. Also, wP vaccines present a longer duration of protection than aP vaccines, which positively influences their effectiveness [54]. At the same time aP vaccines have a great advantage when it comes to reactogenicity. Since they incorporate less types of the bacteria’s protein, they are generally better tolerated and induce less side effects than wP vaccines. Additionally, the side effects they evoke are milder than those corelated with wP vaccine [53]. When it comes to cost, it depends on the awaited impact. Even though aP costs are more up front, it leads to fewer side effect-related healthcare visits and lower indirect costs (lost work time, hospital bills, etc.) compared to wP. When these are factored in, aP becomes more cost-effective overall—particularly in high-income settings. However, when the disease prevention is the priority, wP vaccine becomes more money efficient as they require fewer doses [52].

Comparison of the vaccine types is presented on Table 1.

Multiple vaccines are currently used against *B. pertussis*. In lower-income countries such as Indonesia, India, and parts of Africa, wP vaccines are primarily used due to their lower cost. In contrast, aP vaccines are preferred in higher-income regions like the United States, the United Kingdom, and the European Union because of their improved safety profile [55]. There are multiple vaccines approved by various health-related organizations all around the world.

The most commonly used vaccines are listed and compared by their composition in Table 2.

While acellular pertussis vaccines differ in the number and amount of included antigens, they are unified by their reliance on purified pertussis components and can thus be considered as a single vaccine category in contrast to whole-cell vaccines.

## 6. The Characteristics of Immunological System Response to wP/aP Vaccination

### 6.1. The Immune Response of CD4^+^ T Cells in Humans Is Polarized (Th_1_/Th_17_ vs. Th_2_) Depending on the Form of the Vaccine Administered in Childhood

A number of studies have shown that the immune response profile generated by human PBMCs (newborns and children derived) is dependent on the type of administered vaccine. Exposure to the wP led to the synthesis of INF-γ and IL-2 with low IL-4 and IL-5 expressions. In turn, aP induced the synthesis of INF-γ and IL-4/5. The antigen most effective in inducing IL-4/5 cytokine expression was identified to be PRN [56,57]. Importantly, a number of studies in the human population (in vivo) have confirmed that exposure to the acellular form of the vaccine polarizes the immune response towards Th_2_ and towards Th_1_/Th_17_ in the case of wP [58,59,60]. van der Lee et al. [61] evaluated the efficacy of post-vaccination responses in children of 4 and 9 years of age vaccinated with the combined DTaP or DTwP vaccine (DT stands for diphtheria and tetanus). Despite the higher IgG antibody titers against PRN, FHA, and Ptx in children vaccinated with the cell-free (that is, acellular) form at the age of 6 years, it was observed that children vaccinated with DTwP developed a better humoral response at the age of 9 years. It has been proven that the use of aP vaccine resulted in weaker INF-γ expression but stronger for IL-13 at all checkpoints (4 and 9 years). These observations, similar to the studies above, strongly indicates induction of a Th_2_-dependent response via aP vaccination [61]. Figure 3 presents pathways of immune response to wP vaccine.

In another study by the same authors, it was shown that in pediatric patients pre-vaccinated (2, 3, 4, and 11 months of age) via DTaP or DTwP, subsequent DTaP vaccination (4 years) and Tdap (9 years) resulted in different intensities of humoral response. Children vaccinated from the beginning in the aP regimen at 4 and 9 years of age were characterized by a significantly higher percentage of IgG4 antibodies against all vaccine antigens (correlation with the predominance of Th_2_ responses). In turn, children vaccinated in the pre-vaccinated form of wP, despite the subsequent vaccination with the acellular form, obtained a higher IgG1 titer [62]. This is a valuable note because IgG1-3 antibodies are able to activate the complement and fight pathogens more effectively (unlike IgG4), guaranteeing better immunity via, e.g., antibody-dependent phagocytosis or MAC activation [63]. A number of studies conducted in earlier years confirm the described relationship; the researchers postulate that the primary form of the vaccine, used in childhood, determines the polarization of T lymphocytes in later life [57,64,65]. These reports correlate with previous studies alerting to insufficient immune CD4^+^ T responses in pediatric patients vaccinated only with aP versus those vaccinated (or revaccinated) with wP. Children revaccinated with the second form at the age of 6 years had stronger IL-17 expression than peers vaccinated with aP [59,66]. A series of studies also showed that children pre-vaccinated with DTwP despite taking aP-like doses underwent strong Th_1_/Th_17_-dependent polarization. In turn, patients vaccinated from the very beginning with the cell-free form were characterized by the predominance of Th_2_ cells [67,68,69]. da Silva et al. showed that regardless of the antigenic stimulation model (aP or wP), subsequent exposure of T cells (in vitro) to cytokines INF-β or TGF-β induced an increase in IL-4, IL-9 and INF-γ secretion (stimulation of INF-β) by naive CD4^+^ cells and had no effect on the cytokine profile after TGF-β exposure [70]. Strong evidence for the presented concept is provided in the study by Sheridan et al. [71]. Researchers have shown that in pediatric patients vaccinated with different vaccines (wP, aP, and mixed cycles involving both forms in different orders), a higher rate of pertussis occurs (per 100,000 people/year) in children pre-vaccinated with DTaP than wP. The study may illustrate how in practice the immunological polarization described above may affect susceptibility to disease.

Reports from the above-mentioned studies clearly indicate that the form of the vaccine obtained in childhood may affect susceptibility to the disease in the future. It has been shown that while acellular vaccine may generate high (~68.8%) vaccine efficacy in the protection against disease development in adolescents during the first year of DTaP administration; however, >4 years after immunization from the last dose, there is a decrease to <9% protection [72]. A meta-analysis by McGirr et al. indicated that over time (regardless of the three- or five-dose regimen) only about 10% pediatric patients will remain immune to pertussis after ~8.5 years from the last dose [73]. de Celles et al. indicate in their study (pediatric population 5–9 years) that the efficacy of aP vaccine (protection against disease development) in children exceeds 75%; however, after 5 years after administration of the last dose of DTaP, efficacy drops to about 65% [74]. The usage of the five-dose schedule of DTaP vaccine resulted in a gradual increase in the incidence of pertussis among children born in the years 1998–2003 in the USA (Minnesota and Oregon) within 6 years from the last dose. In Minnesota, there was an increase from 15.6 to 138.4/100,000 patients within 6 years of vaccination [75]. Importantly, it was shown that vaccination with the use of the series of 5/6 doses of aP in childhood gives lower immunity stability in pediatric patients than the same series preceded by at least one additional dose of wP. The relative risk of the five-dose scheme and the six-dose scheme when compared to patients who received ≥1 dose of wP was established at 6.76 and 2.46, respectively. These reports clearly indicate that immunization with wP significantly, although not completely, reduces the risk of disease occurrence [76]. In this chapter, we also described the phenomenon of immune imprinting—shaping a specific profile of the vaccine response (Th_1_ vs. Th_2_) in patients depending on the type of the originally used aP or wP vaccine [59,61,62,66]. As with blunting, the phenomenon limits the effectiveness of the later-life immune response. In the case of vaccination with aP, Th_2_ polarization occurs depending on a limited spectrum of antibodies (the advantage of IgG4) [77]. However, it has not been reported whether the aP vaccine provides insufficient immunity in the pediatric population. A retrospective cohort study conducted in Panama assessed the effectiveness of vaccination with a hexavalent aP vaccine and booster vaccination with wP in the pediatric population (infants 6.5–18.5 months of age and children 18.5–72.5 months of age). The described scheme was characterized by relative and absolute effectiveness of >95% in the prevention of disease development [78].

### 6.2. The Absence of B. pertussis Carriers on the Nasal Mucosa During Vaccination with wP Is Possible Due to Local Proliferation of T_RM_ Cells and Shaping Local Immune Memory

Exposure to the aP vaccine was associated with an increased population of terminally differentiated CD4^+^ T cells than with wP. However, the high dose of *B. pertussis* antigens in the cell-free vaccine, combined with the high percentage of mentioned T cells, is associated with a shortened vaccine efficacy. The described process may be reflected in the lower effectiveness of the aP vaccine. A number of studies have noted that adolescents vaccinated with aP have a limited ability to long-term T-memory cell proliferation compared to wP. This led to a decreased ability to form cytokines upon repeated exposure to *B. pertussis* antigens and, consequently, impaired the post-vaccine response [79,80,81]. Many in vivo studies, conducted using various animal models, provided interesting observations on this term.

#### 6.2.1. Murine Model of Infection

Ross et al. and Brummelman et al. [20,82] have shown that exposure of mice to primary *Bordetella* infection or vaccination with wP generates a Th_1_ and Th_17_-dependent immune response, when aP poorly recruited Th_1_ cells with a predominance of Th_2_ or Th_17_ responses. In an in vivo study using knock-out mice for the gene encoding IL-4 and IL-17A, it was shown that the defect of the IL-4 gene did not condition a decrease in inflammatory cell recruitment and IgG synthesis, after exposure to aP. In the same animals with IL-17A gene defect, the response after vaccination was not sufficient. Hence, the conclusion that a complete immune response requires Th_1_ stimulation (INF-γ mediated) rather than Th_2_ in the case of infection, which occurs with, for example, vaccination with wP [20,82]. Exposure to the aP vaccine was associated with an increased population of terminally differentiated CD4^+^ T cells than with wP. However, the high dose of *B. pertussis* antigens in the cell-free vaccine, combined with the high percentage of the mentioned T cells, is associated with a shortened vaccine efficacy. The described process may be reflected in the lower effectiveness of the aP vaccine. A number of studies have noted that adolescents vaccinated with aP have a limited long-term T-memory cell proliferation ability compared to wP. This led to a decreased ability to form cytokines upon repeated exposure to *B. pertussis* antigens and consequently impaired the post-vaccine response [83,84,85]. Wilk et al. [86] have shown that T resident memory (T_RM_ CD69^+^, CD103^+^) cells allow mice to develop immunity against *B. pertussis* despite blocking the ability of T cells to migrate to lymph nodes (used fingolimod). Polarization of the response towards Th_1_/Th_17_ was observed and, additionally, the presence of the described cells, despite the inhibition of the sphingosine-1-phosphate pathway, conditioned the development of immunity against the pathogen. The described process was confirmed via the transfer of the mentioned T cells to the lungs of naive mice for exposure to the bacterium [86]. Given the similarity of the immune mechanisms present in the primary infection and vaccination with wP, Misiak et al. [87] evaluated the potential role of the Tγδ lymphocytes in immunity against *B. pertussis*. These cells have been shown to induce the early synthesis of IL-17 and Rorγt stimulating the Th_17_/Th_1_ response. Researchers also identified a specific phenotype (T CD27^−^, CD44^+^ Vγ4γδ) of lymphocytes proliferating in the lungs of mice, specifically responding to bacterial antigens. These cells were then locally proliferated and converted into T_RM_ memory cells [87]. The reports of Misiak et al. correlate with the results of other researchers [79,80]. Wilk et al. [81] in another study showed that C57BL/6 mice vaccinated with aP and wP were differently protected against bacterial colonization. While aP provided protection in the lungs, wP also protected the nose area. Moreover, the presence of T_RM_ cells secreting INF-γ and IL-17A in the nasal mucosa was only demonstrated in the vaccination with wP. When fingolimod was used during immunization and in previously immunized mice, limited efficacy of vaccination with wP was demonstrated in the first group, while in the second group, strong expression of IL-17 was noted. Further analysis showed that the response in this group was not impaired due to T_RM_ memory cell activity [81]. Holubova et al. [88] investigated the consequences of vaccinating BALB/c mice with extra antigens (FIM2/3, LOS, dACT) with a cell-free vaccine. It was assessed whether the modification could induce the ability of aP vaccine (1/20 of baseline concentration) to interfere with mucosal colonization by *B. pertussis*. It has been shown that the presence of adjuvants that combated bacterial occupancy in vivo in the lungs of mice, while it promotes colonization of the nasal mucosa. Interestingly, the extensive aP vaccine further induced polarization of the Th_2_ response, with IgG1 antibody synthesis (high titre) and slight IgG2 formation. In turn, expression of TNF-α and INF-γ was initially higher in apparently vaccinated mice (alum + 1/160 of the exit dilution) than in mice vaccinated with aP enriched with adjuvants [88]. Figure 4 presents pathways of immune response to aP vaccine.

#### 6.2.2. Baboons Model of Infection

In a series of studies by Warfel et al., [89,90,91,92,93,94,95] in a population of baboons (vaccinated with both aP and wP) a whole-cell vaccine was shown to induce protection against severe progression of disease with faster eradication of pathogens (colonization of nasopharynx) than in vaccinated with aP and unvaccinated organisms. Moreover, the acellular vaccine did not prevent the transmission of *B. pertussis* and did not affect the colonization of mucous membranes as opposed to wP. At the cellular level, wP vaccine has been shown to induce a Th_1_/Th_17_-dependent response similar to the original contact with bacteria in unvaccinated subjects. Meanwhile, the aP vaccine determined the Th_1_/Th_2_ response, which was suboptimal and did not lead to effective infection prevention [89,90,91,92,93,94,95].

### 6.3. Mechanisms of Innate Immunity Against B. pertussis Play Role as Factors Critically Limiting Th_1_/Th_17_—Dependent Polarization of Lymphocytes

According to Higgins et, al, a defect in the TLR encoding gene resulted in immunodeficiency in mice when stimulated with the wP vaccine in vivo. The TLR pathway itself was responsible for the induction of IL-12, TNF-α, and IL-17 expression during vaccination, which enabled the activation of Th_1_ and Th_17_ subpopulations [96]. Allen et al. [97] demonstrated the significant efficacy of intranasal immunization with aP vaccine enriched with adjuvants: c-di-GMP (cyclic diguanylate) and TLR2 agonist. The specificity of the adjuvants used allowed the polarization of the post-vaccine response to shift towards Th_1_ and Th_17_ at the site of the standard-induced Th_2_ dependent response. Particularly important is the immunization by stimulated local proliferation of resident memory Th_17_ cells in the nasal mucosa area, which in the long run provide protection against colonization and transmission of *B. pertussis* [97]. Dunne et al. [98] have shown that a number of newly discovered *B. pertussis* lipoproteins strongly induce the TLR2 pathway and consequently stimulate the Th_1_ lymphocytes response and IL-17A synthesis. The synthetically developed analog of one of the lipopeptides (LP1569) was added as an adjuvant to the aP vaccine (in vivo murine model). The combination resulted in the potentiating of the Th_1_/_17_ response (in place of Th_2_) and the intensification of the synthesis of specific IgG2a antibodies. Further analysis also demonstrated the increase in the secretion of INF-γ and IL-12/17 via CD4^+^ T cells [98]. Borkner et al. [99] demonstrated in an in vivo *B. pertussis* infection model in mice that IL-17A synthesis present in the primary T cell response induced T_RM_ (CD69^+^, CD103^+^) cells and Siglec-F^+^ neutrophils proliferation in the nasal mucosa. The described population of neutrophils is characterized by an increased ability of phagocytosis, ROS generation, and NET formation. In an in vivo model using knock-out mice for the IL-17A gene, expression of said interleukin has been shown to be a critical limiting factor for the described immune phenomena. Knock-out individuals were characterized by a reduced percentage of Siglec-F^+^ neutrophils and an increased bacterial load in the nasal mucosa [99].

### 6.4. Plasma Cell Activity and Immunoglobulin Synthesis Profile Are Determined by the Form of Vaccination

Vaccination with wP was shown to be more effective than aP regimen in inducing T cell proliferation in draining lymph nodes in CD1 mice, building-up of the T_FH_ (T follicular helper cells) cell population, and consequently increased CXCL13 secretion. This resulted in an increase in the percentage of B memory cells and increased plasma cell activity [100]. Valeri et al. reported that, in mice, initial immunization scheme (aP or wP) defines the polarization of the T-response independently of the subsequent revaccination scheme; this B cell response is the result of the immune booster used [77]. The primary immunization resulted in a stronger proliferation of the germinal center B cells in the case of wP, causing a stronger proliferation of memory B cells and a wide spectrum of antibodies to be secreted: IgG1, IgG2b/c, and IgG3 (with only IgG1 and 2b in aP). A booster vaccination for the wP vaccine also resulted in a stronger systemic response (at the level of lungs, lymph nodes, bone marrow, and spleen) than the aP form. It has been reported that mice pre-vaccinated with an aP regimen, after switching to wP, improved the quality of the immune response, comparable to individuals from the beginning who were vaccinated with wP. Moreover, Valeri et al. [77] indicated that stronger stimulation of memory B cells in the case of the wP vaccine results from a greater response of T_FH_ cells to this form of vaccination. Increased secretion of CXCL13 was reported, resulting in increased proliferation of B cells in murine germinal centers of lymph nodes and, subsequently, more effective formation of memory B cells than after aP stimulation [94]. Other researchers have reported a predominant amount of IgG4 in response to an acellular vaccine [101]. Raven et al., in turn, proved that in DTwP vaccinated mice, Th_1_ response with increased synthesis of opsonins from the IgG2a/b/c and IgG1 group dominated [102]. Another study reported that in different age groups (primarily vaccinated with aP or wP), a booster dose of aP in the majority of cases induced a stronger proliferation of IgG1^+^ plasma cells than when vaccinated in the past with wP. Moreover, the response of plasma cells to the vaccine (synthesis of IgG1 and IgA1) was correlated with baseline IgG titers against PRN, FHA, and Ptx [103]. Significant reports on immune phenomena associated with pertussis vaccine come from a series of studies on live attenuated BPZE1 vaccine [104,105,106]. In an in vivo model (baboon), after intranasal and intratracheal application of the preparation, increased IgG antibody formation against PRN, FHA, Ptx, as well as SIgA [104]. Solans et al. [105] demonstrated in an in vivo mouse model that BPZE1 (as opposed to aP) induced increased expression of SIgA at the nasal mucosa and stimulated proliferation of tissue-resident Th_17_ lymphocytes (CD4^+^, CD69^+^, CD103^+^). The result was effective protection against nasal colonization by *B. pertussis*. At the level of blood serum, high titers of IgA, IgG2a/b, and IgG3 antibodies were recorded, which proved polarization towards Th_1_ [105]. BPZE1 has also been studied in humans in a clinical study conducted by Lin et al. [106]. At the immunological level, all recipients of the live vaccine developed an analogous response profile to the mouse (Th_1_ predominance, IgG1 and IgG3 synthesis). Similarly, the issue of high IgA that was antigenically specific was presented. Notably, patients treated with aP vaccine without exception achieved a Th_2_-polarized IgG synthesis response profile in grades 1–4 with a predominance for IgG2/4. Better expressed synthesis of opsonin antibodies at BPZE1 vaccination resulted in subsequent increased (relative to aP) phagocyte activity [106]. A study conducted in Peru on a group of pediatric patients examined the molecular basis of immune processes occurring during both vaccination with aP and wP (scheme: 2, 4, 6, and 18 months of age) [107]. Researchers indicated an increase in the activity of the TLR-dependent signaling pathways, FcγR-dependent phagocytosis in PBMCs, and increased expression of IRAK-1 and IL-1β molecules in cells isolated from those vaccinated using wP in relation to aP. Importantly, patients receiving the wP vaccine had higher anti-PT antibody titers (153 UI vs. 63 UI, day 150 of the study), while the anti-FHA antibody titer was higher in aP recipients (138.5 UI vs. 17.2 UI, day 150 of the study). On day 150 of the study, the presence of a higher percentage of PT-specific B cells in the wP group was also demonstrated (0.112% vs. 0.045%), and the difference persisted after the end of the vaccination cycle (0.65% vs. 0.1%). No such relationship was noted for FHA-specific B cells.

## 7. Possible Reasons for Increasing Pertussis Prevalence Worldwide

### Phenotypic and Genotypic Changes in B. pertussis-Limitation Factors of aP Vaccine Efficiency

One of the reasons for the decrease in the effectiveness of the acellular vaccine is the appearance of the *B. pertussis* strain characterized by the presence of a new allele-*ptxP3* promoter of the gene encoding Ptx [108]. This strain is characterized by an extremely high expression of bacterial toxin in contrast to the PtxP1-dominant strain before the introduction of the aP vaccine. In a series of studies conducted over the last 20 years, it was shown that the increasing prevalence of the new strain correlated with an increase in the number of pathogen-related infections [109,110,111,112,113]. de Gouw et al. [114] performed a complex comparative analysis of gene expression of key virulence factors of *B. pertussis* and showed a number of differences between their profile in the PtxP1 and PtxP3 strains. The new, more virulent strain was characterized by increased expression of the *fim3* gene as well as *lpxE*-encoding an enzyme that modifies the LPS structure. Moreover, the genes encoding Vag8 transporters (the role shown in interfering with esterase of complement C1 component) and BpaC were also expressed. In addition, there was an increase in the expression of highly virulent T3SS (type 3 secretion system) proteins, including *btcA* and *bteA*. There was also an increased expression of the genes encoding pertussis toxin: *ptxB-D* [114,115]. Importantly, in a study conducted on the Iranian population in the years 2005–2017, bacteria of the PtxP3 strain, which were additionally characterized by a lack of FHA expression, were detected [116]. The PtxP3 strain was also analyzed in a proteomic study by Luu et al. [117]. Researchers reported that *B. pertussis* L1423 was characterized by increased expression of *prn-2* and *tcfA* genes. Increased synthesis of pertactin and the colonization factor of the tracheal epithelium type A led to an improvement in the adhesion capacity of the strain. In turn, a reduced expression of T3SS proteins (BopN, BopD, BteA, and Bsp22), Vag8 and BipA was observed, which could lead to reduced immunogenicity of the pathogen. At the genome level, a significant mutation of the *bscI* gene was observed, which led to impaired expression of T3SS proteins [117].

It is presumed that aP vaccines contributed to selection pressure, promoting the survival of bacteria with reduced expression of strongly immunogenic antigens [29,118,119,120]. Reduced expression of the pertactin-encoding gene (*prn*) as well as the presence of new gene alleles, e.g., *prn-2* in *B. pertussis* shows a correlation with the increase in prevalence of vaccinations using the acellular vaccine [108,121,122,123,124,125,126]. Payne et al. showed that out of 6540 collected *B. pertussis* isolates, as many as 45.3% of them presented the PRN^(−)^ phenotype [127]. This is a growing problem, because PRN is the main antigen of the aP vaccine, which strongly induces the formation of immunoglobulins conditioning the bactericidal effect [128]. Importantly, mentioned selection pressure leads to the appearance of bacterial phenotypes that are negative for the expression of more than one immunogenic virulence factor. Both negative isolates for the formation of PRN and Ptx as well as PRN and FHA were described [124,129]. An important factor limiting the effectiveness of the acellular vaccine also seems to be the multitude of bacterial fimbriae variants. Mutations of the *fim2* and *fim3* genes reduced the effectiveness of aP vaccines based on this antigen [130,131]. The predominance of the mutant *fim3B* allele in bacterial isolates has been a major cause of pertussis rebirth in the USA in recent years [132]. Selective pressure also led to an increase in the occurrence of *B. pertussis* isolates with reduced expression of predominant after the introduction of aP vaccine alleles: *fim3-1* and *fim3-2* [133]. The Bart et al. [131] analysis of the collected *B. pertussis* strains confirmed high antigenic variability in the FIM3 system, apart from typical vaccine antigens (fim3-1, fim3-2), fim3-3 and fim3-6 were detected. The results indicate a potential risk of loss of vaccine efficacy through high variability of *B. pertussis* fimbriae [131]. Bouchez et al. reported changes in the *B. pertussis* PtxP3 genome due to the deletion of the *fim2* gene promoter (loss of FIM2 fimbriae) and allelic polymorphism, leading to the expression of both *fim3-1* and *fim3-2* [134]. In a study conducted with isolates collected during the epidemic in Brazil in 2014–2018, a significant share of the PtxP3/Fim3-24 strain (94/136 isolates) was demonstrated. Importantly, due to the long duration of use of the vaccine wP in this country, researchers estimated the possibility of selection pressure induced via this vaccine (a commonly used vaccine strain [Ptx P1/Fim 3-1] Bp-136) [135]. The effects of the appearance of the new *ptxP3* allele of the Ptx-encoding promoter are shown in Figure 5.

Zeddeman et al. [136] reported about another phenomenon that may contribute to a decrease in the effectiveness of aP vaccines. Researchers showed that DTaP vaccine induced in PRN^(−)^ bacteria phase-change loss of FHA expression (murine model in vivo). At the molecular level, a frameshift mutation of the *fhaB* gene was identified. Loss of bacterial expression of PRN and FHA leads to an inefficiency of aP vaccination in animals [136]. The expression of the *fhaB* gene can be inactivated by changes in the homopolymer pathway G (10 Gs to 11 Gs), which leads to the formation of a shortened FhaB protein. Zeddeman’s reports in this respect are consistent with the results of other researchers [124,137,138]. Weigand et al. in their study showed that among the 722 collected isolates of *B. pertussis*, there were both non-expressing PRN and FHA and a strain characterized by reduced PRN and Ptx formation [124]. As in the case of changes in the pertactin gene area, this phenomenon is also attributed to the influence of selection pressure caused by years of vaccination with aP [139]. The increased expression of Vag8 reported by the researchers is one of the important features of the PtxP3 strain. The Vag8 autotransporter inactivates esterase of complement C1 component, which limits the effectiveness of the initial phase of response against bacteria [111,140,141]. Interestingly, *B. pertussis* is able to induce suppression of the inflammatory response mediated by dendritic cells and macrophages during infection [142]. In the study, bacterial isolates from both aP and wP vaccinated patients were incubated with HEK-Blue cells. Selective activation of TLR2 and NOD2 receptors and induction of IL-10 synthesis resulted in cross-linking of the PD-L1 molecule on the surface of leukocytes (in vitro). Importantly, it was noted that the PtxP3 strain induced the TLR2 and NOD2 response pathway and IL-10 synthesis to a higher degree than that present before the introduction of the PtxP1 vaccine [142]. There was no significant difference in the generation of suppression of inflammatory response between PRN^(−)^ and PRN^(+)^ strains [142]. In the face of these reports, it is noteworthy to mention the induction of IL-10 expression mediated by one of the components of T3SS- BopN protein [143]. In addition to this ability, BopN can interfere with NF-κB and MAPK signaling in immune cells. It has been shown that the inhibition of IL-10 expression may increase the effectiveness of vaccination against bacteria; hence, further research on the described structures seems to be important [144].

Numerous researchers raise the subject of the influence of the main virulence regulator of *B. pertussis* two-component regulators of the BvgAS system as a factor that may influence the avoidance of the immune response by the pathogen [145,146]. The regulator is characterized by a three-phase expression: Bvg^(+)^, Bvg^(−)^, and Bvg intermediate [147]. The last of these phases are characterized by the expression of virulence factors such as FHA or FIM [148]. Transcriptomic analysis of collected PtxP1 and PtxP3 strains showed increased expression of *bvgS* and *bvgA* genes in the P3 strain. In addition, increased expression of Ptx-encoding genes, *ptxA-E*, was correlated with this. In an in vivo mouse model, PtxP3 was shown to have an increased ability to colonize the nasal mucosa, which was most likely due to increased virulence [149]. Significant reports also come from the study by Karataev et al., which showed that *B. pertussis* mutants Bvg^(−)^ presented constantly in the avirulent phase, developed prolonged colonization of the lower respiratory tract (in vivo model, macaques) [150]. It has been shown that single nucleotide substitutions in the *bvgA* gene can lead to RNA polymerase-binding disorders with the gene promoter in *Bordetella*. As a result, both a decrease and increase in *fhaB* gene expression were observed [151]. Hiramatsu et al. described isolate *B. pertussis* derived from a 5-month-old infant, the pathogen was triple negative for synthesis: FHA, Ptx, and PRN. Molecular analysis showed the presence of a substitution mutation in the area of a single nucleotide that resulted in the exclusion of the function of the BvgS protein [152]. Taking into account the reports from the previously cited research, this phenomenon, present in the environment, can promote the evolution of negative FHA strains [124,136]. Reduced immunogenicity conditioned by the modified expression of the BvgAS system requires further research to better understand the mechanism by which it may interfere with the effectiveness of aP vaccination.

The mentioned genotypic changes in *B. pertussis*, the formation of which is largely attributed to the selection pressure caused by the appearance of the aP vaccine, constitute a serious epidemiological problem. Variable expression of virulence factors may result in reduced immunogenicity, which, combined with the ability to modulate the immune response (among others via IL-10), hinders the proper functioning of immune mechanisms against the pathogen. Additionally, disturbing the non-specific immune response (complement system) may facilitate the colonization of the respiratory tract and cause the so-called silent infections, which promotes bacterial transmission in the population.

## 8. Vaccine-Dependent Factors Diminish Its Sufficiency

### 8.1. Silent Infections Promotes Asymptomatic Colonization and Spreading of Pertussis in Populations

The phenomenon of asymptomatic *B. pertussis* infections and associated transmission of the pathogen in the environment also seems to be an important problem. It may be particularly dangerous for age groups with higher exposure to the more severe course of infection due to the activity of the immune system (newborns, infants, and the elderly) [153]. The transmission of the pathogen in the household area can be completely asymptomatic, which is a concern in this regard, according to reports from a series of studies [154,155,156,157,158,159,160]. The phenomenon, according to recent studies, is driven by the commonly used aP vaccine, which is not able to induce an immune response, limiting the possibility of the colonization of mucous membranes and, consequently, leading to silent infections [91]. The Warfel et al. [95] study in an in vivo model (baboon) showed that vaccination with wP or aP generated high titers of antibodies against bacterial antigens (the highest in the case of aP). However, the ability to induce the formation of antibodies made it possible to combat severe infection (lack of cough, leukocytosis) but not moderate colonization [95]. From the findings of later studies, it is already known that Th_2_-dependent responses induced by aP vaccines are not able to sufficiently induce T_RM_ cell proliferation in the mucosal region. This results in exposure to the colonization of, for example, the nose or nasopharynx and consequently the risk of asymptomatic transmission of *B. pertussis* [81,86,87]. An interesting solution to this problem seems to be the currently studied intranasal vaccines, which not only recruit Th_1_/Th_17_ cells but also stimulate the immunity of mucous membranes (T_RM_ and the synthesis of SIgA) [161,162,163].

### 8.2. Original Antigenic Sin Phenomena May Decrease Efficacy of aP Vaccine

Another reason for impaired aP vaccine efficiency is found in the phenomenon of original antigenic sin, also called linked epitope suppression [164,165]. The phenomenon involves generating increasing amounts of memory B cells that are highly specific to vaccine antigens in response to repeated doses of the aP vaccine. This results in this cell population competing with naive B cells and leads to a decrease in the ability of the immune system to recognize bacteria with modified antigenic composition. A similar phenomenon has been described in the context of, among others, vaccination against the influenza [166]. In the context of the aP vaccine, the basis for this phenomenon is seen in the changed structure of PT epitopes undergoing chemical detoxification. As a result, antibodies that are more specific to the vaccine component than toxins naturally formed by the pathogen are formed based on such antigens [167].

### 8.3. Blunting Phenomenon Leads to Impaired Immunity Against Pertussis in Paediatric Populations

The issue of increased incidence of pertussis is particularly noticeable in the population of newborns and children <1 years of age. They are also the groups most exposed to the more severe course of infection [22]. In the case of newborns, it is recommended to start vaccination during pregnancy to ensure immunity from maternal antibodies until the child receives the first dose [168]. Numerous researchers have shown that vaccination during pregnancy significantly reduces the incidence of disease in children too young to receive the first dose of DTaP vaccine [168,169,170,171,172]. However, aP vaccine administered during pregnancy generates a phenomenon called blunting, which involves neutralizing *Bordetella* antigens by antibodies from the maternal penetrating placenta. As a result, the child postnatally has a weaker immune response to the pathogen due to suboptimal stimulation of B cells [173,174]. The research conducted by Knuutila et al. [175] showed that the blunting phenomenon occurring in children was positively correlated with the mother’s initially high antibody levels during pregnancy. The process itself translated into appropriately reduced levels of antibodies and plasma B cells (excluding PT-specific ones) in newborn children [175]. Voysey et al. [176] showed in a meta-analysis on this issue that exposure during pregnancy to the aP vaccine generated in later life a weakened synthesis of antibodies against Ptx, FHA (11% relative to the control group), and PRN (22%). Importantly, after the initiation of the aP vaccination course in the described pediatric patients, the above-mentioned IgG synthesis disorder persisted up to 24 months of age [176]. Another study showed that vaccination with wP form in infants born to vaccinated and unvaccinated mothers (via Tdap) during pregnancy resulted in different antibody titers 1 month after the first dose of vaccine at birth. Newborns of vaccinated mothers had significantly lower levels of IgG anti-Ptx, -FHA, and -PRN. After one month, the phenomenon persisted further in the IgG group for Ptx and FHA. The group of infants born to unvaccinated mothers had a better ability to synthesize IgG anti-Ptx. Despite the originally improved functionality of antibodies present in children vaccinated with aP, the booster dose reversed this effect in favor of wP [177]. A meta-analysis by Abu-Ray et al. [178] found a similar relationship. Antibody levels (after primary vaccination) against Ptx, PRN, FHA, and FIM2/3 were significantly lower in children of women vaccinated during pregnancy with Tdap than in unvaccinated women. Importantly, this phenomenon also persisted in the context of IgG anti-FHA and -FIM2/3 after the use of a booster vaccine [178]. Sapuan et al. [179] demonstrated that vaccination of pregnant women using either 3 or 5 valent DTaP vaccine (combined with polio IPV vaccine) does not generate differences in antibody titers to pertussis antigens in their children at birth. However, after the primary vaccination, children in the control group (mothers not vaccinated during pregnancy via Tdap) achieved higher titers of IgG anti-Ptx. Importantly, after the administration of a booster dose in preschool age, this difference was blurred. Nevertheless, the fact remains that the immune response to *Bordetella* was weakened to ~3.5 years of age in children of mothers vaccinated with the aP vaccine during pregnancy [179]. However, whether the described phenomenon in a real way affects effective immunity against pertussis in later life is debatable and requires in-depth research.

## 9. Antibiotic Resistance Among *Bordetella pertussis* Strains

Antibiotic resistance to *B. pertussis* is reported all over the world, but the most disturbing results are described in mainland China, where resistance to macrolides ranges from 70 to 100%. In previous years, *B. pertussis* resistance was associated with the PtxP1 line, and recently obtained PtxP3 isolates from China also show resistance [180]. One of the first-line antibiotics used in the treatment of whooping cough are erythromycin (ERY), azithromycin, and clarithromycin belonging to the macrolide group; alternatively, in treatment not including macrolides, trimethoprim/sulfamethoxazole is used in a dosage depending on the patient’s age. The optimal time to start therapy is the first 3 weeks, referred to as the catarrhal stage. Treatment with drugs such as antihistamines, corticosteroids, bronchodilators, and antitoxins is not recommended due to the lack of appropriate testing [181]. The dose of ERY is 40–50 mg/24 h in children administered every 6 h for 14 days, while for adults it is 2 g/24 h administered every 6 h. Azithromycin for children is dosed at 10 mg/kg of body weight on the first day, then 5 mg/kg of body weight once between day two and five. For adults, 500 mg on the first day and 250 mg between day two and five. Clarithromycin is administered at a dose of 15–20 mg/kg body weight per 24 h in two divided doses for 7 days for children, and for adults, it is 1 g/24 h in two doses for a week [182]. The first resistant strain was detected in 1994 in Arizona, USA, where the MIC was >64 μg/mL. Known mechanisms of resistance to macrolides include a mutation in the 23S rRNA gene of the bacterial ribosome, and the acquisition of methylase genes, which are resistant to ERY (*erm*), as well as the expression of the MexAB-OprM efflux pump [183,184]. *B. pertussis* resistance to erythromycin was examined, which was associated with a mutation in the 2047 S.C. sequence setting (Sanger Center) in the 23S RNA gene in the microorganism’s ribosomes (*rrn*). In the tested strain of *B. pertussis* Tohama 1 in the Sanger Center, three copies of the 23S gene were observed, which indicates three copies of the rrn operon. This may be the reason for genetic heterogeneity, whereby only a strain with multiple copies of mutations in this gene may become resistant to the given macrolide. The PCR-RFLP test has been suggested to detect strains resistant to ERY [185].

The second type of resistance involves the addition of a methyl group in the 23S RNA of the bacterial ribosome, which leads to blocking of the ERY binding site, which is associated with the acquisition of resistance by acquiring the ERY-resistant methylase gene. However, *B. pertussis* has been shown to lack this gene, and no case has been identified in which this resistance mechanism has been shown to be activated, so this possibility remains theoretical [180]. A study has also been conducted that suggests the existence of another resistance mechanism. Since the *erm*, *mef*, and *ere* genes were shown to be absent, the presence of the mexAB-oprM system, which could act via an efflux pump that would remove the hydrophobic ERY molecule from the bacterial cell, was also tested. This mechanism is active in *P. aeruginosa* and occurs in *Bordetella* spp. However, a deletion has been observed in the mexA and oprM genes in *B. pertussis* that prevents the operation of this pump [184]. In 2024, an increase in pertussis cases was observed in France, Denmark, the Czech Republic, and Spain, although macrolide resistance remains rare in France. Macrolide-resistant *B. pertussis* shows genetic variability and individual phylogenetic groups are listed, such as Bp-AgST4 (alleles *ptx P3*, *fhaB 1*), Bp-AgST8 (alleles: *ptx P1*, *fhaB 1*), and Bp-AgST37 (alleles *ptx P1*, *fhaB 3*). One resistant strain detected in France in the period 1 January–31 May 2024 with genotype Bp-AgST4 remains related to strains from China [22]. A study conducted in Shanghai, China in the years 2016–2022 showed a significant increase in MRBP from 36.4% to 97.2%. The genetic profile of the occurrence of MT195 ptxP1/prn1/fhaB3—MRBP has also changed, replacing the MT28 ptxP3/prn2/fhaB1—MRBP strain in 2020 [186]. Trimethoprim/sulfamethoxazole continues to be highly effective in combating *B. pertussis* infections. Additionally, levofloxacin or doxycycline can be used in adults, while beta-lactams are used in children. However, the above antibiotics are less preferred than macrolides, which achieve better concentrations in the respiratory tract [187].

## 10. Discussion

Pertussis poses a significant challenge to modern medicine: despite widespread vaccination, many regions of the world are seeing an increase in cases. This phenomenon highlights the complex relationship between host immunity, pathogen adaptability, and public health strategy. The most at-risk groups remain the most vulnerable: infants and the elderly. In the context of infants, especially premature babies (born before 31 weeks of gestation), it has been shown that vaccination (both aP and wP) leads to an insufficient immune response [188]. An impaired response to the vaccine has been demonstrated in both CD4^+^ and CD8^+^ T cells [189]. Similarly, in the case of B cells (decreased T cell–B cell interaction, fewer germinal centers) [174]. In a number of children, it was noted that even after completion of vaccination with a booster, the concentration of antibodies against bacterial antigens remained low [190]. In the context of older patients, it is postulated that the decrease in response to pertussis despite vaccination is due to impaired T and B cell function. While it is widely known that T cell reactivity generally reduces with age, memory B cells have been shown to have a similar ability to synthesize IgG against PT as in young people [191]. The difficulty in producing high-affinity antibodies (e.g., IgG1) against bacterial antigens such as FHA in older patients remains a problem [192]. Undoubtedly, the problem of establishing a correlate of protection against *B. pertussis* also contributes to this. Multidimensional protection (different endpoints for disease symptoms vs. colonization vs. transmission) is necessary, which is not provided by currently available vaccines. Consequently, the lack of an internationally accepted correlate contributes to the inability to achieve an adequate immune response. In addition, for many infectious diseases, we have studies confirming a specific level of neutralizing antibodies or T cells that provide lifelong protection against infection, as is the case with some viral diseases [193]. However, the situation is different in the case of pertussis, as this has not been established despite many years of research. Due to its variability in this regard, pertussis still requires research and increased vigilance on the part of clinicians.

The key issue of the choice of vaccine type remains. Despite their higher reactogenicity, wP vaccines induce a broad and long-lasting immune response, with a predominance of Th_1_/Th_17_ lymphocytes and effective memory cell formation, which provides better protection against both infection and transmission [56,57,58,59,60,62,71]. In contrast, aP vaccines, currently dominant in highly developed countries, generate mainly a Th_2_ humoral response. This results in relatively weaker long-term immunity, reduced protection against asymptomatic colonization of the respiratory tract, and, consequently, the occurrence of so-called silent infections [72,73,74,75,76,86,87]. Immunological phenomena such as original antigenic sin, blunting, or lack of tissue-recruitment of T_RM_ further limit the effectiveness of aP vaccines and contribute to the persistence of bacteria in the population despite a high vaccination rate [86,87,177,178,179,194]. Whereas original antigenic sin and its role are not yet fully understood and require further research, the phenomenon of blunting has been known for a long time and its actual impact on the effectiveness of pertussis vaccination is disputed [195]. For example, a study conducted in Great Britain assessed the effectiveness of vaccination with TdaP in pregnancy over 3 years after its introduction and did not show an increase in the incidence of pertussis in the population of children of mothers vaccinated during pregnancy using TdaP. The vaccine simultaneously prevented infant deaths from the disease at 95% [196].

The remarkable adaptability of *B. pertussis* is no less important. Genetic and phenotypic changes, including the emergence of the ptxP3 lineage, PRN-deficient strains, and fimbriae variability, reduce the immunogenicity of circulating bacteria and diminish the effectiveness of existing vaccines. It appears that the long-term use of aP vaccines has exerted selective pressure, accelerating these evolutionary processes [197]. An important problem in the epidemiological context is the phenotypic variability of *Bordetella* in the context of, among others, PRN synthesis. In recent years, there has been a significant increase in the percentage of strains that do not express PRN, which was attributed to the selection pressure caused by the aP vaccine. For example, in the Netherlands, the share of PRN^(−)^ strains increased from 15% to 25% over the period 2015–2020 [198,199]. However, in the years after the COVID-19 pandemic (2023–2024), pertussis epidemics were observed, caused by Bordetella isolates that expressed PRN in 93% of cases [200]. Importantly, the above-mentioned study indicated that in the post-COVID-19 era, the dominance of strains expressing the *ptxP1* variant of the PT gene promoter was observed, whereas *ptxP3* strains dominated in the Netherlands before the pandemic. The study suggests that the pathogen is constantly evolving, adapting to epidemiological trends. During the COVID-19 pandemic, probably as a result of the disruption of vaccination schedules or reduced exposure of children to whooping cough due to isolation, the dominance of strains that do not form PRN was reversed in favor of those that express it [13].

At the same time, resistance of *B. pertussis* to macrolides is increasingly observed, particularly in China, but also sporadically in Europe, which further complicates epidemic control. Mutations in the 23S rRNA gene, responsible for resistance, limit the effectiveness of first-line therapy and require constant microbiological surveillance [180]. The development of antibiotic resistance among *Bordetella* spp. also involves mutations in the *erm* or *rrn* genes [180,181,182,183]. Antibiotic resistance, combined with prolonged colonization of the mucous membranes present in aP-vaccinated patients (insufficient T_RM_ stimulation), may pose a potential risk of spreading resistant bacteria even in populations with a high vaccination rate. This raises the question of whether we are at risk of epidemics of pertussis resistant to macrolide antibiotics, a topic that requires further research.

These observations raise a number of controversies and unanswered questions. Although animal models (mice, baboons) provide important immunological data, they only partially reflect clinical reality. The described model is difficult to clearly assess the benefits of a given form of vaccination, because no studies on baboons have proven a significant advantage of a given vaccine in preventing colonization and transmission of the pathogen to a satisfactory extent. The mentioned study revealed that it took approximately 17.5 days after infection to combat infection in animals vaccinated with wP, and the use of aP required twice the time. In both cases, it is, therefore, difficult to talk about a satisfactory target effect, which is the effective prevention of colonization and, consequently, bacterial transmission in relation to both forms of vaccination [89,90,91,92,93,94,95]. Epidemiological data from low-income countries, where whole-cell vaccines continue to dominate, are lacking. There is also no consensus on the optimal vaccination strategy, particularly regarding prime-boost schedules and booster dose schedules [201,202].

## 11. Conclusions

Pertussis, despite long-term vaccination programs, remains a significant threat to public health. The biggest challenge is the limitations of cell-free vaccines, which do not provide satisfactory lasting protection or prevent asymptomatic infections. At the same time, the adaptability of *B. pertussis* and the increasing resistance to antibiotics further complicate effective disease control. Future strategies should focus on the development of new generations of (particularly intranasal and mucosal) vaccines and on monitoring pathogen evolution and resistance patterns on a global scale. Only a combination of innovative immunoprophylactic solutions with systematic epidemiological surveillance can ensure lasting disease reduction and protection of the most vulnerable population groups.

It is necessary to develop a studies base that includes, in particular, monitoring changes in the *B. pertussis* genome on a global scale as well as the design of new-generation vaccines that induce a strong cellular response and those that will revise the composition of the target vaccine, enriching it with other antigens, i.e., BrkA or TcfA. Oral and intranasal vaccines have become the latest trend in infection prevention. In the case of *B. pertussis*, the development of mucosal vaccines, limiting the colonization and transmission of the bacteria, seems to be a very important endeavor for researchers to undertake. In addition, expanding research to evaluate a prime–boost strategy combining both existing types of vaccines to extend the duration of protection seems to be the key to success.

## Figures and Tables

**Figure 1 ijms-26-09607-f001:**
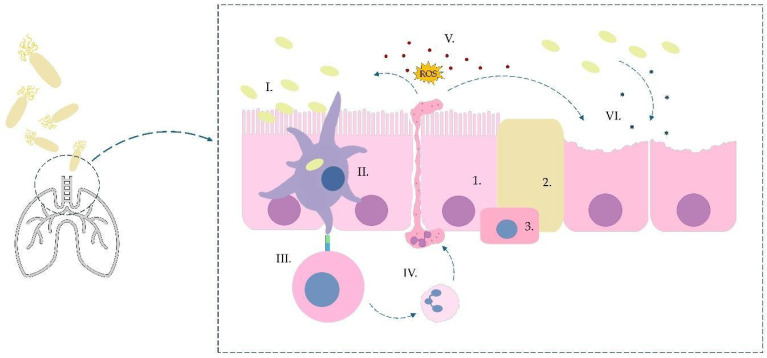
Pathogenesis of *B. pertussis* infection. The infection occurs through droplets, *B. pertussis* colonizes the tracheal epithelium (cilliated cells (1), goblet cells (2), basal cells (3)) using specific adhesins: FHA, PRN and fimbriae (I). Bacteria induce a response from TLR receptors (toll-like receptors), developing an initial immune response. During this time, they are also absorbed by dendritic cells (II), which activate T lymphocytes in the lymph nodes, which differentiate towards Th17 (III). These lymphocytes, via secreted cytokines, activate and cause the migration of neutrophils to the site of infection (IV). During inflammation, numerous antipathogenic factors and reactive oxygen species (ROS) are secreted (V). Bacteria produce numerous exotoxins (e.g., pertussis toxin, cytotoxin), which, together with inflammatory factors, cause massive destruction of the tracheal epithelium (VI).

**Figure 2 ijms-26-09607-f002:**
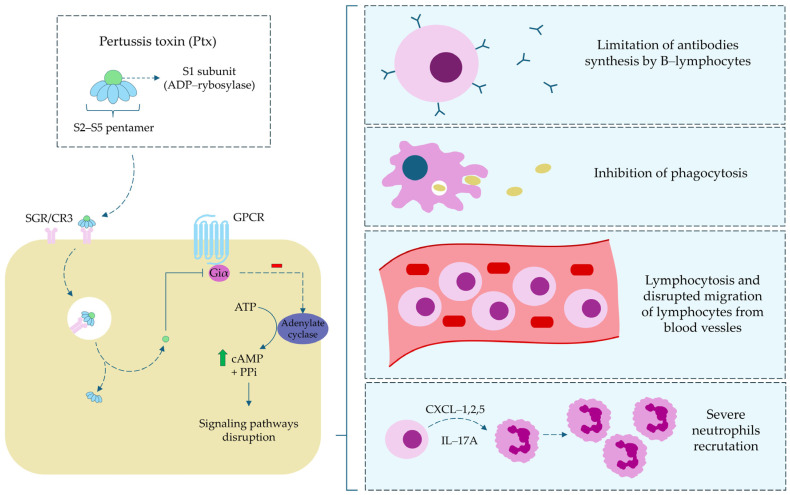
The pattern of action of pertussis toxin (Ptx) with the resulting immunological consequences. Abbreviations: ATP—adenosine triphosphate, cAMP—cyclic adenosine monophosphate, CR3—macrophage-1 antigen, CXCL-C—X-C chemokine ligand, Giα—Gi protein alpha subunit, GPCR—G-protein coupled receptor, IL—interleukin, PPi—pyrophosphate, SGR—sialoglycoprotein.

**Figure 3 ijms-26-09607-f003:**
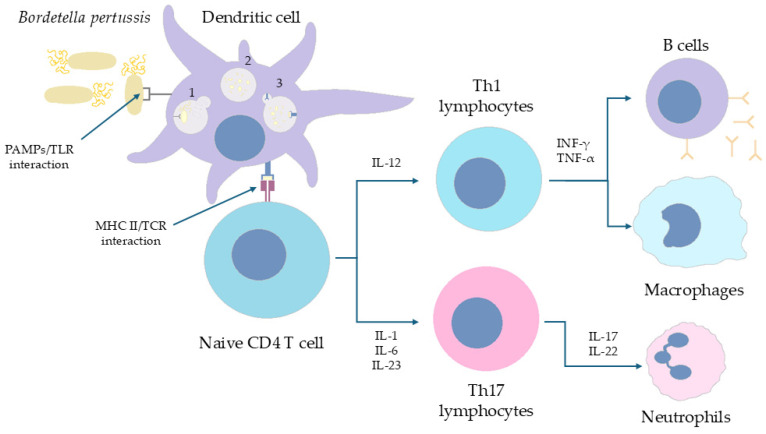
Immunological phenomenon after whole cell anti-pertussis vaccine administration (simplified scheme). Numbers 1–3 presents the sequence of molecular action in antigen presenting cell after PAMP/PRR/TLR interaction: 1—internalization of bacterial pathogen/antigens inside endosome and its fusion with lysosome, 2—enzymatic lysis, 3—fusion of endo-lysosome with vesicle transporting MHC II receptor and exposition of antigen-MHC II complex on cell membrane. Abbreviations: IL—interleukin; INF-γ—interferon gamma; MHC II/TCR—T cells receptor (TCR) recognition on major histocompatibility complex class II (MHC II); PAMPs/TLR—pathogen-associated molecular patterns (PAMPs) recognition by toll-like receptor (TLR).

**Figure 4 ijms-26-09607-f004:**
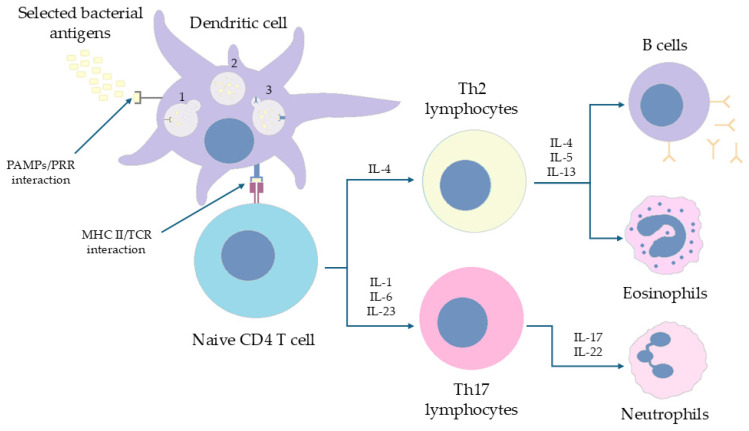
Immunological phenomena after acellular anti-pertussis vaccine administration (simplified scheme). Numbers 1–3 presents the sequence of molecular action in antigen presenting cell after PAMP/PRR/TLR interaction: 1—internalization of bacterial pathogen/antigens inside endosome and its fusion with lysosome, 2—enzymatic lysis, 3—fusion of endo-lysosome with vesicle transporting MHC II receptor and exposition of antigen-MHC II complex on cell membrane. Abbreviations: IL—interleukin; PAMPs—pathogen-associated molecular patterns; PRR—pattern recognition receptors.

**Figure 5 ijms-26-09607-f005:**
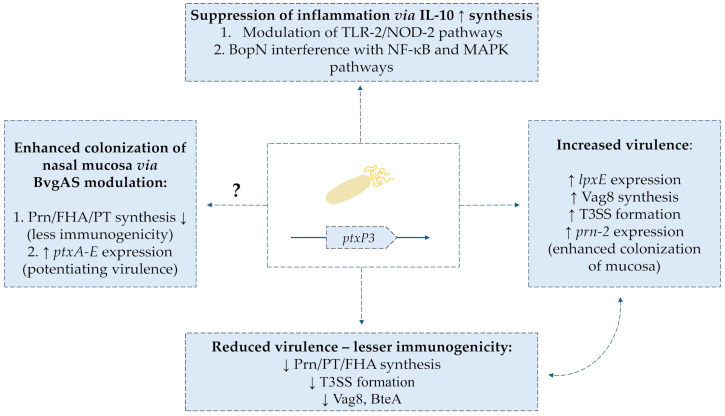
The effects of the appearance of the new *ptxP3* allele of the Ptx-encoding promoter. Abbreviations: BteA—gene encoding *Bordetella* type three secretion system effector A, BvgAS—two-component regulatory system of *B. pertussis* virulence, down arrow—reduced effect/action/expression, FHA—filamentous haemagglutinin, IL—interleukin, *lpxE*—gene encoding a lipid A 1-phosphatase, MAPK—mitogen activated protein kinase, NF-κB—nuclear factor kappa-B, NOD—nucleotide-binding oligomerization domain-like receptors, PT—pertussis toxin, *ptx*—gene encoding pertussis toxin, TLR—toll-like receptor, T3SS—type 3 secretion system, up arrow—increased effect/action/expression, Vag-8—Virulence Associated Gene 8, ?-expected effect.

**Table 1 ijms-26-09607-t001:** Comparison of wP and aP vaccines.

Type of Vaccine	wP Vaccine	aP Vaccine
**Efficacy and effectiveness**	better protection even against colonization	poorer protection
**Duration of protection**	longer	shorter
**Reactogenicity**	higher	lower
**Cost**	more cost-effective when disease prevention is the priority	more cost-effective, when healthcare visits related to vaccines are included
**References**	[52,54]	

**Table 2 ijms-26-09607-t002:** Composition comparison of selected anti-pertussis vaccines.

Name of the Vaccine	Producer	Composition
**Whole-cell pertussis vaccines (wP)**		
Quintanrix	GlaxoSmithKline, London, UK	Inactivated *B. pertussis* strain—not less than 4 international units.
**Acellular pertussis vaccines (aP)**		
Infantrix; Boostrix; Pediatrix; Kinrix	GlaxoSmithKline, London, UK	Always contain detoxified Ptx. May include FHA, PRN, fimbriae (not uniform, but generally present in combinations).
Daptacel; Adacel; Quadracel; Pentacel; Pediacel; Repevax; Hexacima	Sanofi Pasteur Inc., Paris, France	
Vaxelis	Merck & Co. Inc., Rahway, NJ, USA and Sanofi Pasteur Inc., Paris, France	

## Data Availability

No new data were created or analyzed in this study. Data sharing is not applicable to this article.

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
