# Peer review of "Pertussis—A Re-Emerging Threat Despite Immunization: An Analysis of Vaccine Effectiveness and Antibiotic Resistance"

_ijms, 2025, doi:10.3390/ijms26199607_

Round 1

Reviewer 1 Report

Comments and Suggestions for Authors

Please find the comments and suggestions for authors as an attachment.

Author Response

Dear Reviewer 1

We would like to thank you for such a careful review of our manuscript. We appreciate the time you have invested and your intention to improve our article. Below, we present our position on the individual points raised by the reviewer.

  1. We would like to thank you for your suggestion. We have added it in the place we consider most appropriate. In order not to hinder the reception of the Abstract, we have decided to include this information in the Introduction. It reads as follows:

„It is also worth noting that a clear correlation with protection against Bordetella pertussis has not yet been established, although elevated levels of some pertussis-specific antibodies (e.g., anti-PT-IgG, anti-PRN) are associated with a reduced risk of disease. However, studies show that antibody titers decline over time, and IgG-PT alone is not sufficient to predict protection.. This uncertainty likely contributes to occasional vaccination failures and the need for booster doses [25,26]”.

  1. We would like to thank the reviewer for this comment. We had absolutely no intention of being impartial. The aim of our review was to present the problem from the following perspective: despite the availability of such and such vaccines on the market, the problem of whooping cough is still alive and well, it has returned. And we absolutely do not want to comment on which vaccine is better, as we do not feel qualified to do so. However, we emphasize in our statements that transmission should be interrupted, as this will stop the deadly danger to infants.

From our point of view, for the purposes of this manuscript, a general description of the difference between vaccines („Whole-cell vaccines (wP) promote immune polarization toward Th1 and Th17 subsets, while acellular vaccines (aP) favour a Th2 response, which is more effective at reducing carriage rather than preventing infection [19]”) is sufficient. Delving into the details would be important if the topic were approached from an immunological perspective. But that is a topic for a completely separate article.

Furthermore, our point of view has been clearly stated, as we demonstrate that the most effective vaccination strategy is one that includes both vaccines simultaneously:

„The aim of the pertussis vaccination is to resolve long-term T-cell responses. Immunization was designed to activate primarily Th1 and Th17 cells, which play a critical role in clearing pathogens from the respiratory mucosa and limiting B. pertussis transmission. Whole-cell vaccines (wP) promote immune polarization toward Th1 and Th17 subsets, while acellular vaccines (aP) favour a Th2 response, which is more effective at reducing carriage rather than preventing infection [19]. Moreover, aP vaccines elicit limited cellular immunity (mainly enhance the synthesis of antibodies) and only modest Th1 activation, thereby offering suboptimal long-term protection against B. pertussis [19,20]. Consequently, an effective vaccination strategy that ensures both infection control and environmental bacterial reduction should ideally include both wP and aP components”.

  1. Thank you for pointing out our inaccurate wording. We have changed the previous sentences to the following:

„Several reasons for this phenomenon can be mentioned, but what is particularly important from the microbiological point of view is the correlation of the increased number of pertussis cases with the introduction of a new form of vaccine – the acellular vaccine  in place of the whole-cell vaccine „.

And

„Although the introduction of acellular pertussis (aP) vaccines is often cited as a contributing factor to the resurgence of pertussis, other factors should also be noted.
The aforementioned asymptomatic course of disease, often occurring in adults, plays an important role, making them a difficult-to-detect source of pertussis spread [9].
Furthermore, improvements in diagnostic methods (such as the increased use of PCR), greater clinician awareness, and lower testing thresholds can also been identified as contributing factors to the increase in reported cases [31]. Continuous changes in vaccination schedules, including the timing and number of booster doses administered, are also important, as demonstrated by modeling and a serological study by Paireau et al. in France in 2013, following a change in the childhood vaccination schedule in that country [32]”.

  1. We would like to thank you for pointing out this inaccuracy. All abbreviations have been standardized, and their explanations appear only when they are first used. In addition, all abbreviations from the Abstract have been removed.
  2. As suggested by the reviewer, we have added information about vaccinations in Poland. We have also added an appendix containing the current vaccination schedule used in Poland:

„The only exception is Poland, where up to 18 months of age four doses of the whole-cell vaccine (DTP) are administered, unless there are contraindicated (if they occur, then the acellular form is administered). Subsequent vaccinations for Polish children are based on the acellular vaccine (DTaP, Tdap)”.

  1. We have changed the previous sentence to the following:

“According to WHO data from 2014,, 50 million cases of whooping cough are recorded each year, 300 thousand of which end in death”.

  1. For clarity, we have added a paragraph referring to data from the Koch Institute. This data relates to the period after the COVID pandemic, referring to the period before that (2013):

“For example, in Germany, according to data from the Robert Koch-Institut (RKI),  25,271 pertussis cases were officially recorded in 2024 compared to 3,429 cases in the preceding year—indicating a more than sevenfold increase and the highest incidence since the introduction of national mandatory reporting in 2013 [14]”.

  1. Thank you for bringing this to our attention. We have added the necessary information.
  2. Thank you very much for pointing this out. Of course, the incorrect implication came from our statement. Of course, the introduction of these vaccinations in the 1950s and 1960s was not a modification of T's response. However, referring to point 5 of this Review, this passage has been changed, and we hope that the reception of this part of our Article is no longer misleading:

„In order to protect against pertussis, common and in many countries also mandatory vaccination of children was introduced (Poland – 1960; German Democratic Republic – 1964, Federal Republic of Germany – 1969; Great Britain – 1950s, USA – registration of the first whole-cell pertussis vaccine in 1914, DTP vaccine commonly used in most states since the 1980s) Currently, the vaccination schedules of most of the above-mentioned countries use the acellular form of the pertussis vaccine, primarily as a combination vaccine with C. diphtheriae and C. tetani antigens (DTaP) [16–18]. The only exception is Poland, where up to 18 months of age four doses of the whole-cell vaccine (DTP) are administered, unless there are contraindicated (if they occur, then the acellular form is administered). Subsequent vaccinations for Polish children are based on the acellular vaccine (DTaP, Tdap)”

  1. In accordance with the Reviewer's other suggestions, this sentence (after slight modification) is in the right place. It provides a link between the previous part of the Manuscript and the part that follows it:

„Despite the calm for many years, recently there has been an upward trend in cases with cyclical waves every 3-5 years, with a peak in the summer months [22,23]”.

  1. We have slightly changed the wording of this statement, following the reviewer's suggestion. However, the suggestion: „It would be valuable if distinctions between infection and symptoms could be made in applicable places throughout the paper to emphasize the message of the review. E.g., aP vaccines prevent symptoms very well but not that well colonization” is not clear to us. We have slightly changed the wording of this statement in accordance with the reviewer's suggestion. However, the suggestion is not clear to us. Our passage clearly states that aP provides greater colonization reduction than wP. Therefore, the booster dose provides immunity because it guarantees a sufficiently high level of antibodies. In addition, the sentence we added in response to reviewer no. 1's suggestion supplemented our statement. We hope that in its current form, the message is clear and not misleading:

„For the reasons mentioned above, it follows neither vaccination in childhood nor recovering from the disease guarantees lifelong protection. Therefore, it is possible and quite common to be infected with pertussis several times in one’s  life; unless a booster dose of the vaccine is administered every 5 years, which allows pertussis-specific antibodies and cellular immunity to remain above pre-booster levels”.

  1. This timeframe is very vague, as there is insufficient scientific data. Therefore, in order not to mislead the reader, we have not addressed or elaborated on this issue.
  2. We would like to thank you for this comment, which we would like to discuss a little. We do not understand the reviewer's position as to why, in the case of the dominance of strains lacking the ability to produce PRN, this virulence factor should be excluded from the virulence factors of this bacterium. It is known that when a bacterium has adhesive factors, the disease develops much faster and more easily. However, bacteria without this ability are also pathogenic, as they have other virulence factors, such as toxins. In this case, the stage initiating the infection is omitted, which does not mean that such a strain is less or more important or infectious. In our opinion, the wording we have used is entirely correct.

14) Dear Reviewer, thank you very much for your detailed feedback. We agree that presenting exact data on T-helper responses, antibody levels, and precise efficacy percentages would provide a more quantitative perspective. However, the aim of this table is to provide a broad synthesis of vaccine-induced immune responses and protection mechanisms rather than an exhaustive compilation of numerical data. Including all specific estimates and age-stratified reactogenicity data would significantly expand the scope and length of the manuscript, potentially obscuring the overarching messages we wish to convey. We have therefore chosen to retain a more concise summary while highlighting the key protective effects of the vaccines.

15) Dear Reviewer, thank you very much for your thoughtful feedback and for highlighting important aspects of our manuscript. We fully agree that focusing on adaptive changes in B. pertussis is central to the aim of this paper. At the same time, we believe that providing a more detailed description of the clinical course of the disease is valuable, as it helps to place these adaptive changes in a broader biological and medical context. Our intention was to offer readers, including those less familiar with pertussis, a more comprehensive understanding of how clinical manifestations relate to pathogen adaptation and vaccine efficacy. For this reason, we chose to retain this section, though we have revised it for conciseness and clarity. We hope this rationale addresses your concern and demonstrates the relevance of this content to the overall goal of the review.

16) We appreciate the suggestion to shorten Table 2. Since the key point of our review is to analyze aPs as a distinct class of vaccines, we have simplified the table to highlight only their common features. We acknowledge minor variations exist between products, but for clarity and consistency in the discussion, we have chosen to present them here as a homogenous group.

17) Dear reviewer, thank you for your notice. We corrected this spelling mistake.

18) Thank you for this comment, dear reviewer, in fact the conclusions we presented regarding the baboon model could have been too far-reaching. In the discussion section, we included a fragment devoted to the limitations and interpretation of the outcomes resulting from them.

19) Thank you for this comment, the indicated literature item was included in the revised version of the manuscript.

20) Thank you for your comment, we carefully revised the manuscript and removed unnecessary abbreviations and repetitions.

21) Thank you for drawing attention to this issue. As You may have noticed, the structure of our review is not only about aspects devoted to immunological issues and the molecular effects of specific forms of the vaccine (aP vs wP). The review also included extensive sections devoted to genotypic changes caused by selection pressure among Bordetella strains or the aspect of increasing antibiotic resistance. Our goal was to comprehensively develop the topic, both in terms of immunological phenomena and those mentioned above. Due to the structure of the article we have adopted, developing the indicated issues would require a significant expansion of these sections and the creation of an excessively long article. The issues you mentioned are important and we will certainly develop them in the future in another article directly focused on the issue of humoral response. However, in order not to leave this comment unanswered, we have introduced extensions to topics devoted to FHT-cells, memory B-cells or the issue of the impact of age on the response to vaccination.

23) Dear reviewer, the way we actually presented the topic may have seemed a bit controversial, which is why the structure of the chapter has been improved. It has been expanded with a section devoted to the phenomenon of original antigenic sin. Reports from retrospective studies have been moved to another chapter.

25) Dear reviewer, as you suggested, we have modified this subchapter and quoted the literature items indication. In the discussion section, we tried to discuss the issue from the perspective of the real effectiveness of vaccinations, see the Amirthalingam et al 2016.

26) Thank you for this comment. As I mentioned above, the article discusses a number of factors that directly or indirectly contribute to the increase in pertussis prevalence. I am afraid that it is difficult to estimate on how increasing antibiotic resistance affects the effectiveness of pertussis vaccination. When we included this section in the article, we thought more about the cause-effect sequence: bacteria resistant to antibiotics recommended for the treatment of pertussis proliferate on the mucous membranes of the respiratory system, and this prolonged colonization in asymptomatic patients may promote the transmission of bacteria in the population. In this way, in addition to the phenomenon of vaccination-induced selection pressure (decrease in immunogenicity), antibiotic resistance also seems to be a factor contributing to difficult control of pertussis.

27) Thank you for this comment. In line with the recommendation, the structure of the conclusions was changed in favor of a short summary, while some of the threads previously included in it were moved to the discussion section. We hope that these changes have eliminated the feeling of bias regarding the form of vaccines.

In accordance with the Reviewer's suggestions, our Review has been significantly improved. We invite the Reviewer to familiarize themselves with the revised version of the Manuscript. We would like to thank the Reviewer for their thorough analysis of our Article. The suggested constructive changes have undoubtedly contributed to the creation of a version that is more accessible to the reader. The changes that were not implemented, due to their different nature than that presented in our Article, motivated us to write another Review, which we hope will be published soon. We hope that the work we have put into improving this Article will meet with the Reviewer's approval and satisfaction. 

Best wishes

Anna Duda-Madej

Reviewer 2 Report

Comments and Suggestions for Authors The manuscript reviews the main factors contributing to the resurgence of pertussis, highlighting among them the reduced efficacy of vaccination and comparing the immune responses induced by whole-cell and acellular vaccines. It also highlights that phenotypic variation may be associated with decreased vaccine effectiveness and emphasizes antimicrobial resistance to certain classes of antibiotics, such as macrolides. Abstract- should be more concise and objective, considering it covers several themes related to the increased incidence of pertussis, often without a direct correlation between the topics. The purpose of the abstract is to provide the reader with a clear and concise preview of the article's content. However, the current abstract is 22 lines long, which compromises its objectivity. The body of the article (review) consists of an extensive introduction (20 pages) and a brief conclusion (only one page). Various topics are addressed in detail and are well-referenced, covering aspects such as vaccine-induced immune responses (both whole-cell and acellular), phenotypic variability, and increasing antibiotic resistance, among others. However, for a review article, the approach must be easy to understand and written in a clear manner, without compromising scientific rigor. In several sections, the reading resembles the structure of book chapters, which may hinder comprehension. Regarding the comparison between whole-cell and acellular vaccines, there is a stronger focus on developed countries that use the acellular vaccine. I suggest including more data on the response to the whole-cell vaccine, such as evidence from Latin America. I also recommend a more detailed discussion on how the findings relate to public health and contribute to disease control. It was not possible to identify the methodology used for selecting the articles included in the review in the main text. Although methodology is mentioned in the author contributions section ("Author Contributions: Conceptualization, A.D.-M.; methodology, H.B.; J.Ł. a..."), it is important that this information be clearly described in the body of the article. I suggest adding a specific methodology section explaining the inclusion and exclusion criteria used. It is also recommended to include a discussion section that integrates the themes addressed in the introduction (22 pages), such as phenotypic variation, disease resurgence, and the selective pressure generated by whole-cell and acellular vaccines. Although the discussion is well written, it would be pertinent to add an analysis of the clinical and epidemiological impact of these bacterial adaptations in the global context. Regarding the references, it is worth noting the high number of citations, 185 in total, many of which are over ten years old. I suggest prioritizing more recent references, preferably from the last five years, to ensure the review reflects current knowledge and best practices. With regard to the tables and figures, it is recommended that they present appropriate formatting and be proportional to the body of the text. In several sections (pp. 7, 8, and 9), an excessively close arrangement of the tables is observed, which compromises the clarity and proper interpretation of the information. Therefore, a more careful reorganization of these elements within the textual structure is suggested. Conclusion: The topic addressed is extremely relevant, especially given the increasing number of pertussis cases worldwide. I recommend implementing the suggested revisions so that the text more closely resembles a scientific review rather than a book chapter. I recommend that the article be revised accordingly.

Author Response

Reviewer 2

We would like to thank you for such a careful review of our manuscript. We appreciate the time you have invested and your intention to improve our article. Below, we present our position on the individual points raised by the reviewer.

Abstract- should be more concise and objective, considering it covers several themes related to the increased incidence of pertussis, often without a direct correlation between the topics. The purpose of the abstract is to provide the reader with a clear and concise preview of the article's content. However, the current abstract is 22 lines long, which compromises its objectivity.

Thank you for this opinion, dear recenent, the abstract has been corrected and we hope that it is now more objective and coherent.

I suggest including more data on the response to the whole-cell vaccine, such as evidence from Latin America. I also recommend a more detailed discussion on how the findings relate to public health and contribute to disease control.

With reference to this opinion, dear reviewer, we have included in the review appropriate references to research conducted in Peru and Panama to outline the problem of pertussis in a way that is also representative of Latin America. Moreover, in the discussion section we included a number of references to the latest research on the epidemiology of pertussis in the context of selection pressure and changes in trends after the COVID-19 epidemic.

It was not possible to identify the methodology used for selecting the articles included in the review in the main text. Although methodology is mentioned in the author contributions section ("Author Contributions: Conceptualization, A.D.-M.; methodology, H.B.; J.Ł. a..."), it is important that this information be clearly described in the body of the article. I suggest adding a specific methodology section explaining the inclusion and exclusion criteria used.

A methods section has been added, which sets out the criteria for including the publication in the review.

It is also recommended to include a discussion section that integrates the themes addressed in the introduction (22 pages), such as phenotypic variation, disease resurgence, and the selective pressure generated by whole-cell and acellular vaccines. Although the discussion is well written, it would be pertinent to add an analysis of the clinical and epidemiological impact of these bacterial adaptations in the global context. Regarding the references, it is worth noting the high number of citations, 185 in total, many of which are over ten years old. I suggest prioritizing more recent references, preferably from the last five years, to ensure the review reflects current knowledge and best practices. 

With reference to your opinion, dear reviewer, we have added a discussion section in which we tried to include your indication of the issue. A number of cited studies from the last 5 years have been added throughout the article to make it more relevant.

Best wishes

Anna Duda-Madej

Reviewer 3 Report

Comments and Suggestions for Authors

I have finished the review of the manuscript entitled "Pertussis – a Reemerging Threat Despite Immunization: An Analysis of Vaccine Effectiveness and Antibiotic Resistance." it is a very complete review which I read with true interest and consider valuable.

Nevertheless there are certain aspects that need further revision:

1. paragraphs are only a couple of lines, while other are too long. Please consider sub-paragraphs, especially where separate mechanisms or experimental models (e.g., baboons vs mice) are discussed, pages 12-13.

2. The bias in aP vaccines is repeated in several forms (sections 1, 5.1, 5.2, 7.1), also, TRM cells and mucosal immunity are explained multiple times with similar wording. Please avoid repetitions.

3. Topic transition can be improved, particularly from section 6 (genotypic changes) to section 7 (vaccine factors); no connecting paragraph or summary is present.

4. a really important aspect to revise for mantaining a neutral tone is that a lack of neutrality or biased wording can be clearly identified in some statements:

a superiority of wP is implied in wording :

“wP vaccine becomes more profitable...”
“...only the whole-cell vaccine provided protection...”

This way of presenting information seems to overlook or diminish the  safety concerns and public resistance that led to aP adoption.

Wording in presention of causal relation vs correlation:

“The aP vaccine strongly induced the synthesis of IgG antibodies against this antigen… Therefore, it promoted colonization.”
Correlation is not always causation. This conclusion may overstate the direct role of the vaccine in facilitating colonization.

also, by using unsustained adjectives, e.g.: “Forgotten bacterium” – arguably emotive or sensational phrasing for a scientific paper.

Abstract seems like an extended background and introduction, lacking clear,  structure summarizing objectives, methods, and conclusions implict or explicitly.

Use of English is different from one paragraph to another, from coloquial to scientific, e.g. line 229, states "ecchymosis on the patient's face" when correct term would be facial ecchymosis,

Table 1 has typo errors, " Comparison of wP and aP vaccines. Type of vaccine wP vaccine aP vaccine Efficiancy and effectiveness. also, the scale is presented from better to poorer? adjectives could be changed to more appropiate terms.

Subtitle 6 

"Possible reasons for increasing pertussis prevalence worldwide "--- is it prevalence? It should be frequency or incidence, please revise according to the definitions and differences between prevalence and incidence as concepts are not synonims.

"Children revaccinated... had stronger IL-17 expression than peers vaccinated..." (should be consistently past or present).

"antibodies produced after an received vaccination", should be “after a received vaccination”.

Consider changing:

“...post-vaccination response profile, symptomatic of vaccination with aP...” to: “characteristic of vaccination with aP” is clearer.

“Due to their heterogenous nature, wP vaccines provide more effective protection...” to: needs clarification; “heterogeneous” in what sense?

Please, 

Consider performing a comprehensive language editing pass by a native English-speaking scientific editor to enhance clarity and fluency.

Standardize terminology use (e.g., avoid “cell-free” and “acellular” interchangeably without clarification).

I am confident that the revised version will be improved.

Comments on the Quality of English Language

able 1 has typo errors, " Comparison of wP and aP vaccines. Type of vaccine wP vaccine aP vaccine Efficiancy and effectiveness. also, the scale is presented from better to poorer? adjectives could be changed to more appropiate terms.

Use of English is different from one paragraph to another, from coloquial to scientific, e.g. line 229, states "ecchymosis on the patient's face" when correct term would be facial ecchymosis,

paragraph lenght is too variable, some paragraphs are long in excess while others are 2-3 lines.

Author Response

Dear Reviewer 3

We would like to thank you for such a careful review of our manuscript. We appreciate the time you have invested and your intention to improve our article. Below, we present our position on the individual points raised by the reviewer.

1) Dear reviewer, with reference to your comments, we have introduced an appropriate division of experimental models, dividing the section from pages 12-13 into subchapters devoted to mice and baboons.

2) Dear reviewer, abbreviations have become standardized, unnecessary repetitions have been removed as you suggest.

3) As suggested, there is a summary at the end of Chapter 6 that seamlessly introduces the reader to the phenomenon of silent infections and OAS and the topic of blunting.

4) Dear Reviewer, thank you for drawing attention to this important issue. As suggested, we have revised the manuscript to improve the language and have given particular attention to the sections you highlighted. We hope that these changes successfully address your concerns.

Best wishes

Anna Duda-Madej

Round 2

Reviewer 2 Report

Comments and Suggestions for Authors

The authors have made appropriate revisions and provided clear and satisfactory explanations in response to my previous comments. These changes have improved the manuscript. I recommend the revised version for publication in its present form.

Reviewer 3 Report

Comments and Suggestions for Authors

Manuscript is now clearer and can be accepted in its present form.